# A Network Analysis Study on the Structure and Gender Invariance of the Satisfaction with Life Scale among Spanish University Students

**DOI:** 10.3390/healthcare12020237

**Published:** 2024-01-18

**Authors:** Diego Diaz-Milanes, Vanesa Salado, Carmen Santín Vilariño, Montserrat Andrés-Villas, Pedro Juan Pérez-Moreno

**Affiliations:** 1Department of Quantitative Methods, Universidad Loyola Andalucía, 41704 Sevilla, Spain; ddiaz@uloyola.es; 2Institute of Health Research, University of Canberra, Canberra 2617, Australia; 3Department of Experimental Psychology, Faculty of Psychology, University of Seville, 41092 Sevilla, Spain; 4Department of Clinical and Experimental Psychology, University of Huelva, 21007 Huelva, Spain; santin@dpsi.uhu.es (C.S.V.); pedro.perez@dpsi.uhu.es (P.J.P.-M.); 5Department of Social, Developmental and Educational Psychology, University of Huelva, 21007 Huelva, Spain; montserrat.andres@dpsi.uhu.es

**Keywords:** satisfaction with life, psychometrics properties, network analysis, centrality measures, Bayesian network, university students

## Abstract

Introduction: The psychometric properties of the Satisfaction With Life Scale (SWLS) have been evaluated across numerous languages and population groups, primarily from a factor analysis perspective. In some studies, inconsistencies in structural invariance have been identified. Objective: This study aims to analyze the properties and gender invariance of the SWLS from a network analysis perspective. Method: A total of 857 Spanish university students were obtained through a stratified random cluster sampling method in a cross-sectional survey design study. Descriptive analysis of the items, partial-correlation network, Bayesian network model estimation, and invariance analysis by gender were conducted. Results: The instrument did not exhibit any floor or ceiling effects. Each item can be considered univariately normally distributed, and all items clustered in a single and stable community. The partial-correlation network model and centrality measures were stable in the full sample and invariant across genders. Item 3 emerged as the most central node in the network with the highest predictability. The Bayesian network indicated that items 2 and 4 initiate the process, while item 5 acts as the sink, and items 1 and 3 act as mediators. Conclusions: The SWLS can be used as a unidimensional measure, and the total score and relationships among items are stable and reliable. Any potential differences among genders cannot be associated with the functioning of the instrument. The predictability of every item was high, and the Bayesian network clearly identified different roles among the items.

## 1. Introduction

Satisfaction with life has been identified as a primary component of subjective well-being, defined as a cognitive process and an individual’s overall evaluation of one’s life [1]. Current studies associate this component with people’s mental and physical health [2,3,4,5]. For example, Lopes and Nihei [6] demonstrated that university students with higher life satisfaction developed more adaptive management of circumstances and psychological well-being during the COVID-19 pandemic, resulting in greater resistance to depressive and anxiety symptoms and stress.

To assess life satisfaction in the population, Diener et al. [7] developed the Satisfaction With Life Scale (SWLS) as a foundational tool for future scientific work in the field. However, Diener et al. [8] also emphasized the need for psychometrically sound measures of life satisfaction and outlined their characteristics. Subsequently, numerous validation studies on the SWLS have been conducted across diverse population groups, including breast cancer patients [9], outpatient clinic patients [10], individuals with mental disorders [11,12] and informal caregivers [13], among others. These validations have taken place in various countries such as Sweden [12], Lithuania [14], Pakistan [15], Vietnam [16], Korea [17], Brazil [18], Peru [19], Chile [20,21], Mexico [22,23] and Spain [24], among others. This extensive research has led to the SWLS accumulating over 30,000 citations and being translated into, at least, 39 different languages [16,25]. Consequently, it has become the most widely used measure for assessing the life satisfaction component of subjective well-being [26,27].

An overview of these psychometric studies indicates that they have followed the classical test theory approach, conducting factor analysis to assess its validity and internal consistency indicators to assess its reliability, as seen in studies such as [28,29]. Some of these studies also incorporated the Item Response Theory approach [12,19].

In addition, many of these studies have included factorial invariance analysis conducted across gender, age, countries, cultures, marital status, level of education, and clinical and non-clinical samples, among other variables [11,16,19,20,22,29,30,31]. However, discrepancies and inconsistencies were found among different studies regarding the level of invariance reached across genders, age, or culture [32], as was the case with gender in its Spanish version [29,33], translated and validated in a large sample of adolescents [24].

While traditional conceptualizations have their shortcomings, novel approaches may offer alternatives. In this case, network analysis [34] could prove helpful in assessing SWLS psychometric properties. The emerging field of network analysis has demonstrated its utility in various areas of psychology and mental health research [35,36], such as evaluating psychopathological symptoms [37,38], treatment effectiveness [39,40], and personality traits [41,42]. The network approach posits that variable co-occurrence arises not from a common underlying cause, but from variable-to-variable interactions [43,44].

Network analysis aims to offer an alternative to conventional analyses, such as factor analysis, to complement the results provided from a novel perspective and contribute to more robust evidence. Therefore, implementing this new approach and its associated techniques is essential to address the current replication crisis in the field [45].

Furthermore, a recent advancement in network analysis involves estimating the directionality of relationships using Bayesian network models [46]. These models could be instrumental in assessing and constructing psychometric instruments by identifying the role of different items in the network [47].

The goal of the present study was to revalidate the SWLS using network analysis, estimating both non-directed and directed models, and testing the invariance of the measure across genders in a sample of Spanish university students.

## 2. Materials and Methods

### 2.1. Procedure and Participants

The present research is part of the Health Behavior in University (HBU) study, which is based on a cross-sectional survey design. The data used in this study were those collected during the 2018/2019 academic year.

The HBU study adhered to the fundamental ethical principles outlined in the Declaration of Helsinki. It was authorized by the Research Ethics Committee of Huelva Centers (CEI) of the Junta de Andalucía (0846-N-19/P1027/19), and written consent was obtained from all participants.

To recruit participants, stratified random cluster sampling was conducted. The strata, with proportional allocation, were determined based on areas of knowledge. Subjects from the first and third years were randomly selected as clusters until meeting the established quota for each area of knowledge. Inclusion criteria for participation in the study required enrolment in a degree program at the University of Huelva and explicit consent for data processing. Exclusion criteria encompassed overseas students (Erasmus) and minors (<18 years of age).

### 2.2. Measures

The participants completed the Satisfaction with Life Scale (SWLS) [7]. The Spanish version, used in the present study [24], consists of a 5-item scale with response options ranging from 1 to 7, where 1 = “totally disagree” to 7 = “totally agree”. The total score obtained ranges from 5 to 35, with higher values indicating higher levels of satisfaction with life.

### 2.3. Data Analysis

#### 2.3.1. Descriptive Analysis of the Items

First, descriptive analyses were conducted to determine the mean, standard deviation, range, skewness, and kurtosis of the SWLS and its items. Additionally, ceiling and floor effects (when ≥15% of participants scored at the extremes of the scale) were assessed for the total scale score [48,49].

#### 2.3.2. Variable Communities Estimation and Stability

A bootstrap exploratory graph analysis [50] was conducted to estimate the number, structure, and stability of variable communities (dimensions of the network) in terms of their structural consistency. A community of variables is formed in the network when nodes are strongly correlated to each other through edges, allowing for the identification and extraction of these communities along with the items belonging to them. Items were retained in their respective communities only if their stability values were higher than 0.70. Items with lower stability values were excluded because they had the potential to compromise the structural consistency of a dimension [50].

#### 2.3.3. Network Structure Estimation and Stability

The structure of the network for SWLS items was estimated utilizing a partial-correlation network, and the association parameters between all nodes were calculated according to Gaussian Graphical Models (GGM) [34]. In Gaussian Graphical Models, if two variables (nodes) are connected in the outcome graph, they are related to each other after controlling for all other variables. Next, we used the least absolute shrinkage and selection operator (LASSO) regression model to set small correlations to zero [51]. This process employed a regularization rule that reports edges cautiously to ensure the simplicity and accuracy of the network structure [52]. The accuracy of the obtained edge weights was assessed by calculating confidence intervals (e.g., 95% CI) for their estimates [34,53].

#### 2.3.4. Centrality Estimation and Stability

Centrality measures reveal the importance of any given node in a network, indicating which variables are the most relevant and the role of each in the model. In the present study, three types of centrality indices were computed: strength, closeness, and betweenness. Strength, the most widely used and robust centrality measure, refers to the weighted sum of all edges connected to a particular node. Closeness takes the inverse of the sum of distances from one node to all other nodes in the network, while betweenness quantifies how often one node is in the shortest paths between other nodes [34,54,55]. Each measure was regularized with values ranging from 0 to 1.

Centrality stability is an indication of the reliability of the network centrality indices. It is assessed using the Correlation Stability (CS) coefficient, which indicates the maximum proportion of observations that can be removed while still maintaining a correlation of 0.70 or higher between the original centrality indices and those of the subsets, with 95% certainty [34]. The stability of centrality indices was estimated using a subset bootstrapping procedure, with a total of 1000 bootstrap samples. Centrality indices were considered stable when their values exceeded 0.50 [53].

#### 2.3.5. Network Structure Invariance

To evaluate the invariance of the network across genders, a network comparison test (NCT) was conducted. This test aimed to identify potential differences at three levels: network structure (assuming identical structures in both networks), global strength (assuming identical overall connectivity in both networks), and edge strength (assuming similar strength for all edges) between the female and male groups [56].

#### 2.3.6. Bayesian Network Model

The construction of the Bayesian network (BN) is carried out in two steps. Firstly, the estimation of the directed acyclic graph (DAG) is performed, and secondly, the BN model is fitted and validated with the study dataset. For the DAG estimation phase, the PC Stable algorithm with no restrictions was employed. To ensure stability in the obtained DAG, a total of 200 bootstrap samples were drawn, and only the edges with a strength greater than 0.85 and a direction greater than 0.5 were retained in the final model [47]. To fit and validate the BN model in the second phase of the process, the dataset was subdivided into 10 folds. A routine was implemented in which 90% of the folds were used to train the model, and the remaining 10% were used for testing. This cross-validation routine was repeated until all potential combinations were explored.

#### 2.3.7. Predictability Capacity

The predictability and error in the prediction were assessed using the coefficient of determination (R^2^) and the root-mean-square error (RMSE) for both the partial-correlation and BN models, allowing for obtaining both an absolute and relative measure of the accuracy of the final models. In the partial-correlation, a node’s predictability was determined by considering the remaining nodes in the network as predictors [57]. In the BN model, the predictability of a node (considered a child) was determined exclusively by the directed edges identified during the DAG estimation phase from other nodes (considered parents). The predictability and error for the prediction of each node were calculated as the means of the cross-validation process, which was the second phase in the BN construction. In both models, the predictability of each node was visually depicted by a border around the node in the graphical representation of the network.

For a comprehensive explanation of these statistical indicators for network analysis and the application of BN, consult the following references [43,46,55,58].

All the analyses were performed using R version 4.2.3 (R Core Team 2018). The required packages were psych (Version 2.2.3), EGAnet (Version 2.0.2), qgraph (Version 1.9.5), bootnet (Version 1.5.6), mgm (Version 1.2.14), NetworkComparisonTest (Version 2.2.2), bnlearn (Version 4.9) and caret (Version 6.0.94).

## 3. Results

### 3.1. Characteristics of the Sample

A total of 970 students agreed to participate in the study. After excluding those who did not complete the consent form, minors, age outliers within the university population, and participants/questionnaires with missing values, the final study sample consisted of 857 students. Among them, 74.45% identified as female, and 25.55% as male, falling within the age range of 18 to 26 years (M = 20.65; SD = 2.15). Participants in the current study were distributed across various fields of study: Arts and Humanities (6.42% of the sample), Engineering and Architecture (1.75%), Natural Sciences (2.45%), Health Sciences (40.37%), and, lastly, Social and Legal Sciences, which constituted nearly half of the sample (49.01%).

### 3.2. Descriptive Statistics

Table 1 presents the means, standard deviations, skewness, and kurtosis of the 5-item scale. The items and global score showed univariate skewness and kurtosis within the range of −2 and +2. Due to the score range of the items and the skewness and kurtosis values, they were considered univariately normally distributed. Only 0.35% of participants scored at the lower extreme of the global score, and 1.28% scored at the upper extreme. Therefore, no floor or ceiling effect was identified.

### 3.3. Network Stability Assessment

The bootstrap exploratory graph analysis identified a community of items with 100% stability, and the 95% confidence interval of edges indicated their accuracy and interpretability. Additionally, the strength, closeness, and betweenness of the obtained network were found to be highly stable, with CS coefficients of 0.750, 0.750, and 0.517, respectively. Therefore, the network structure can be considered unidimensional, and its edges and centrality indices are fully reliable.

### 3.4. Network Structure and Centrality

Figure 1A displays the partial-correlation network structure of the SWLS, where all the identified relationships are represented in blue due to the positive correlation among all the nodes. The network density was 80% (8 non-zero edges out of 10 possible edges), with the most robust connections identified among items 3 and 1, 2, and 4, with weights equal to or higher than 0.35 in those cases.

Regarding centrality measures, item 3 exhibits the highest values in every indicator of strength and closeness, followed by items 1 and 4, while items 2 and 5 prove to be the least central nodes in the network. In terms of betweenness, item 3 holds most of the indirect paths among nodes in the network in comparison to the rest of the items (Figure 1B).

### 3.5. Network Comparison

The results of the NCT indicated no statistically significant differences in their network structure (M = 0.130, *p* = 0.649), global strength of the network (S = 0.039, *p* = 0.705), or individual edges (*p* > 0.05 for all cases) between the group of women and men. Therefore, the partial-correlation network can be considered gender invariant.

### 3.6. Bayesian Network (BN)

The BN was employed to predict the probable direction of edges in the networks. The resulting BN identified items 2 and 4 as the primary parents in the network, indicating a higher likelihood of activating the entire network. Conversely, items 1 and 3 play a dual role as both parents and children, forming an indirect pathway for the effect of items 2 and 4 on item 5, which solely serves as a child (see Figure 2).

### 3.7. Predictability

The predictability and error for each item in the network were estimated using partial-correlation and BN models, and the results are presented in Table 2. In both models, item 3 exhibited the highest predictability, while item 5 had the lowest predictability value. Conversely, the error level showed an inverse pattern, with the lowest value for item 3 and the highest for item 5. For the BN model, predictability or error values were not reported for items 2 and 4, as they were not children of any other node in the model.

## 4. Discussion

To the best of our knowledge, this is the first study to apply network analysis, including partial-correlation and BN models, to investigate the properties of the SWLS and assess the invariance of the network structure across genders.

On the one hand, the stability assessment demonstrated strong reliability of the measure in all its aspects, obtaining a unidimensional structure consistent with previous studies based on factor analysis [23,59,60,61]. Regarding the structure and centrality measures, item 3 (“I am satisfied with my life”) emerged as the cornerstone of the instrument. This could be explained as the item serving as a summary of the construct by itself.

On the other hand, the comparison of the two networks between females and males indicates no differences in network structure, the global strength of the network, or individual edges between the two networks. Thus, the model can be considered invariant across genders. In contrast, the Spanish version used in the present study did not show evidence of gender invariance in its previous publications with adolescents using the classical approach [33].

Network analysis represents a step forward in addressing such inconsistencies, as studies have demonstrated the sensitivity of partial-correlation network analysis in detecting differences that may exist [41,62]. Therefore, the network analysis approach seems a suitable alternative for other language versions, such as Quechua [19], or Serbian and Iranian versions [63], which have obtained similar results concerning gender invariance through factor analysis. Furthermore, this approach may shed light on age and cultural inconsistencies commonly found among different versions [32]. Therefore, the use of bootstrap-based methods provides more stable and generalizable communities of variables and network structures, ensuring comparability of results with other target populations or versions of the instrument in further studies.

The BN model replicated the same edges obtained in the partial-correlation network but added a direction, which was potentially causal. It showed that considering the conditions of your life as excellent (item 2), and achieving the important things you want in your life (item 4), will lead you to be closer to your ideal life (item 1) and being satisfied with your life (Item 3). Consequently, you would not want to change any aspects of your life (item 5). The causal inference pathway obtained is crucial from a theoretical perspective, as it elucidates the temporal dimension in the assessment of life satisfaction.

This pathway suggests that to reach a state where altering any aspect of your life is not needed, the person must achieve the previous milestones, such as the remaining items on the scale, among other elements. This evidence highlights the instrument’s capacity to indirectly cover the theoretical requirement of assessing life satisfaction over a specific time span [64].

In terms of predictability, the variance explained is above 0.30 for all cases, with item 3 (“I am satisfied with my life”) having the highest value in both the partial-correlation and BN. Mean predictability values are comparable to those found in network models of other psychological constructs [57]. The results of both network analyses and predictability measures align with the findings of the study by Jovanović [65], where strong correlations were indicated between the single-item measure of life satisfaction and the global score, as well as the rest of the items and related constructs.

The present study has two main limitations. Firstly, it employed a cross-sectional design. While machine learning algorithms can estimate causal pathways in a set of items, a longitudinal design is necessary to verify these relationships and assess the temporal reliability of the model. Implementing this follow-up would allow us to observe whether the history of changes in certain nodes of the network could precede changes in other nodes, offering additional support for causal inference and potentially establishing Granger causality [66]. Secondly, while the data collection process enabled the acquisition of a representative sample of Southern Spanish university students, future research should consider a broader and more heterogeneous national sample to enhance the generalizability of the findings across different sectors of the population. To achieve this, future studies should move from an exploratory use of network analysis to a confirmatory approach to ensure the invariance of the network across population groups. Furthermore, the use of a network approach can be valuable from a salutogenic perspective [67,68] as it opens the door to new research lines exploring the interrelationships among elements of positive psychology constructs related to satisfaction with life. These constructs include the sense of coherence, happiness, empathy, flourishing, optimism, or curiosity [31,69,70,71]. Additionally, it allows for the exploration of connections with concepts from psychopathology, such as alcohol and/or drug abuse, stress, anxiety, or depression [6,65,72,73], covering the full continuum of health [74,75]. Identifying bridge nodes that may act as potential triggers for good or bad mental health, as noted by Opsahl et al. [54], could facilitate the design and testing of tailored interventions for specific population groups.

## 5. Conclusions

To conclude, this study supports the use of the SWLS as a measure of global life satisfaction among Spanish university students. The instrument can be considered a single factor, and the total score and relationships among items are stable and reliable, with no apparent gender-based differences in the functioning of the instrument. The predictability of every item was high, and the BN identified distinct roles among the items. Future psychometric studies should consider employing a network analysis approach to ensure the validity and reliability of their measurements for both the total sample and each subgroup of analysis. Additionally, the inclusion of BNs built by machine learning algorithms can provide a better understanding of the role of each item in the instrument.

## Figures and Tables

**Figure 1 healthcare-12-00237-f001:**
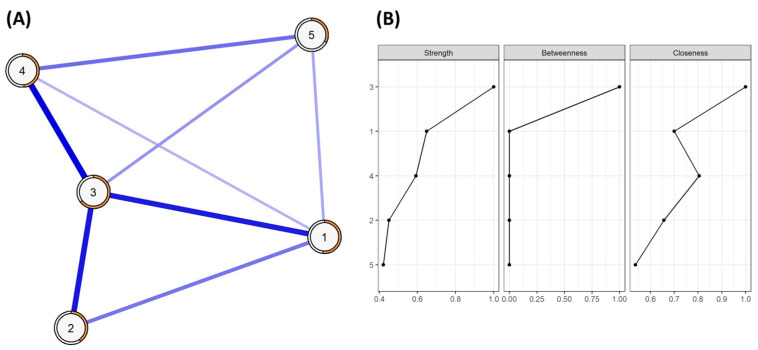
(**A**) Partial-correlation network showing relationships between items of SWLS; (**B**) regularized centrality measures of the network (strength, betweenness, and closeness).

**Figure 2 healthcare-12-00237-f002:**
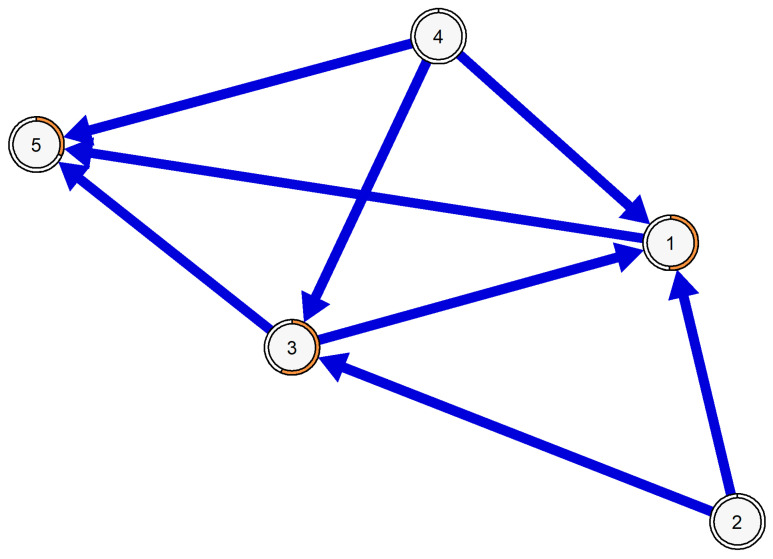
BN model of the SWLS items.

**Table 1 healthcare-12-00237-t001:** Descriptive analysis of each item and the SWLS global score.

Item	Mean	SD	Min.	Max.	Skew	Kurtosis
1. In most ways my life is close to my ideal	4.68	1.30	1	7	−0.63	0.29
2. The conditions of my life are excellent	5.10	1.32	1	7	−0.80	0.57
3. I am satisfied with my life	5.36	1.35	1	7	−1.05	1.18
4. So far I have gotten the important things I want in life	5.33	1.51	1	7	−0.98	0.46
5. If I could live my life over, I would change almost nothing	4.67	1.90	1	7	−0.48	−0.95
Global Score	25.15	5.79	5	35	−0.86	0.67

**Table 2 healthcare-12-00237-t002:** Predictability and error of partial-correlation (GGM) and BN models.

Variable	GGM		BN	
	R2	RMSE	R2	RMSE
Item 1	0.516	0.695	0.508	0.911
Item 2	0.404	0.771	-	-
Item 3	0.638	0.601	0.568	0.882
Item 4	0.479	0.722	-	-
Item 5	0.310	0.830	0.316	1.586

## Data Availability

The data that support the findings of this study are available from the corresponding author upon reasonable request.

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
