# Peer review of "A Network Analysis Study on the Structure and Gender Invariance of the Satisfaction with Life Scale among Spanish University Students"

_healthcare, 2024, doi:10.3390/healthcare12020237_

Round 1

Reviewer 1 Report

Comments and Suggestions for Authors

The submitted manuscript presents a psychometric network analysis for the satisfaction with life scale for Spanish students. The paper has some merit because it might be interesting to see what network analysis has to offer in addition to factor analysis. More discussion regarding this aspect would be warranted.
Detailed comments:
1.    70: Write “classical test theory approach”. There are also numerous other places in the manuscript where words should not be capitalized (e.g., “network analysis”, line 85).
2.    94: It is unclear what is meant by the statement that network analysis provides “robust results”. In particular, why should it be preferred over factor analysis? Is factor analysis thought to be less robust?
3.    Section 2: Use subsections 2.1 and so on.
4.    183: Shouldn’t it read “network comparison test”?
5.    194: write “bootstrap samples”.
6.    203: typo “correlation”
7.    Table 1: Also, present skewness and kurtosis.
8.    Compare the findings of the network analysis with a one-dimensional factor analysis.

Reviewer 2 Report

Comments and Suggestions for Authors

First and foremost, I want to express my appreciation for the opportunity to review your manuscript whose title is to "A Network Analysis Study on the Structural and Gender Invariance of the Satisfaction With Life Scale among Spanish University Students.". I find the topic quite intriguing. This study explores the psychometric properties of the Satisfaction With Life Scale (SWLS) among Spanish university students through a network analysis perspective, investigating its structural characteristics and gender invariance,

To enhance the manuscript's quality, following a thorough examination, I have a few comments and questions for the authors.

- Remove "affiliation" and leave the name of the affiliation.

Introduction:

- Revise the whole document so that the sentences end correctly and do not break syllables. E.g.: pages 2, lines 45, 47.. (and in the title).

- Revise the format of the bibliography is in brackets and with the reference number, not with the names of the authors.

Material and methods

- In the network analysis section, clarity can be improved by explaining some technical terms, especially for readers unfamiliar with network methods.

- In the predictability assessment, it may be beneficial to provide a brief justification of why R2 and RMSE were chosen as assessment metrics.

- In the material and methods section, do not put results. Write the results section, percentage of women and men, the percentage of the fields of study, the percentage of women and men, the percentage of the fields of study, the percentage of women and men

Discussion

- The discussion of limitations could be more detailed. For example, the cross-sectional nature of the design and the need for longitudinal studies to validate causal relationships. The authors could comment on how these limitations might affect the interpretation of your results and how readers should consider these factors when assessing the robustness of your findings.

- The discussion of generalisability of results could be expanded: How might these findings apply to other populations or contexts, and are there possible biases or limitations in the applicability of the results?

References

Ref 2 y 3 incomplete

Round 2

Reviewer 1 Report

Comments and Suggestions for Authors

no further comments